# Predicting Myalgic Encephalomyelitis/Chronic Fatigue Syndrome from Early Symptoms of COVID-19 Infection

Chelsea Hua [1], Jennifer Schwabe [2], Leonard A. Jason [3,*], Jacob Furst [4] and Daniela Raicu [4]

1 Division of Mathematics, Computing, and Statistics, Simmons University, Boston, MA 02115, USA; chelsea.hua02@gmail.com
2 The Grainger College of Engineering, The University of Illinois Urbana-Champaign, Champaign, IL 61801, USA; schwabe6@illinois.edu
3 Center for Community Research, DePaul University, Chicago, IL 60614, USA
4 College of Computing and Digital Media, DePaul University, Chicago, IL 60604, USA; jfurst@cdm.depaul.edu (J.F.); draicu@cdm.depaul.edu (D.R.)
* Correspondence: ljason@depaul.edu

**Abstract:** It is still unclear why certain individuals after viral infections continue to have severe symptoms. We investigated if predicting myalgic encephalomyelitis/chronic fatigue syndrome (ME/CFS) development after contracting COVID-19 is possible by analyzing symptoms from the first two weeks of COVID-19 infection. Using participant responses to the 54-item DePaul Symptom Questionnaire, we built predictive models based on a random forest algorithm using the participants' symptoms from the initial weeks of COVID-19 infection to predict if the participants would go on to meet the criteria for ME/CFS approximately 6 months later. Early symptoms, particularly those assessing post-exertional malaise, did predict the development of ME/CFS, reaching an accuracy of 94.6%. We then investigated a minimal set of eight symptom features that could accurately predict ME/CFS. The feature reduced models reached an accuracy of 93.5%. Our findings indicated that several IOM diagnostic criteria for ME/CFS occurring during the initial weeks after COVID-19 infection predicted Long COVID and the diagnosis of ME/CFS after 6 months.

**Keywords:** Long COVID; PASC; myalgic encephalomyelitis/chronic fatigue syndrome; ME/CFS

## 1. COVID-19 and Long COVID

A recent major threat to global health, the viral infection known as coronavirus disease 2019 (COVID-19) caused by the SARS-CoV-2 virus [1], emerged in December 2019 in Wuhan, China [2]. Since then, it has caused $19 trillion in damages [3] and over 6 million deaths worldwide, and there have been almost 550 million confirmed cases [4]. The high transmissibility and fatality rate [2] of this disease sent many countries into lockdown, requiring citizens to quarantine and wear masks to ensure public safety. Remote work and schooling [5] became commonplace during the pandemic. Beyond the human and financial cost of this pandemic, the pandemic itself and lockdowns have had a heavy toll on individuals in terms of physical and mental health [6].

The symptomatology of the disease itself includes a range of respiratory, gastrointestinal, and vascular symptoms [7]. It affects all ages, but the elderly are particularly at risk, [1] as well as those with underlying health issues, such as diabetes or asthma, in terms of both frequency and severity of the disease. Usually, the acute illness period lasts no longer than 2–3 weeks [1]. However, some patients with COVID-19 have experienced enduring symptoms for months after the original infection [8]. The persistence of COVID-19 symptoms months or even years after infection is referred to as post-acute sequelae of COVID-19 (PASC) or, more commonly, Long COVID [9].

Long COVID symptoms commonly include but are not limited to fatigue (particularly post-exertional malaise or PEM); neurocognitive symptoms (difficulty concentrating,

memory impairment); persistent loss of taste and smell; detrimental effects on multiple organ systems (heart palpitations, breathing issues, chest pain, joint fatigue or pain); and neurological symptoms (anxiety, depression) [8,10,11].

Not only does Long COVID impact a person's physical abilities but it also frequently has detrimental effects on the person's mental health and ability to work [6]. Currently, there is no biological test to diagnose Long COVID or PASC [12]. The diagnosis of Long COVID requires observing a patient's history of COVID, starting from the time of first contraction, and comparing these initial symptoms to current symptoms [10]. The healthcare worker must also examine the patient's medical history to eliminate other possible causes of symptoms, such as underlying diseases or other issues [10]. Diagnosing Long COVID requires a comprehensive medical examination.

Lingering symptomatology is a common phenomenon of COVID, with varying rates of those experiencing at least one long-term symptom [13]. One analysis found the global prevalence of Long COVID to be 37% at 1 month after infection, 25% at 2 months, 32% at 3 months, and 49% at 4 months [14]. A study by Davis et al. [15] found the most common lingering symptoms to be fatigue, PEM, and cognitive dysfunction six months post-COVID. At 7–9 months post-COVID infection, Nehme et al. [16] found that 39% of patients reported continuing symptoms. Another study that followed COVID-19 patients over time found that, after two years, 55% of COVID-19 patients reported at least one lingering symptom [17].

Due to this endurance of symptoms, finding predictors of Long COVID may assist with early detection and symptom management. An investigation by Sudre et al. [18] found evidence suggesting an association between developing Long COVID and experiencing five or more symptoms during the initial first week of contracting the virus. Symptom severity in the initial weeks of COVID-19 infection has also been found to be predictive of a long-term prognosis of Long COVID [18].

Studies have attempted to group symptoms into categories. For example, Kenny et al. [11] used cluster analysis to define three clusters: predominantly pain symptoms, predominantly cardiovascular symptoms, and one cluster with fewer symptoms than the other two. Other studies have also attempted to group symptoms [19,20]; however, these studies were limited by a small sample size. Principal component analyses performed on symptoms from the initial three weeks of infection and ongoing symptoms yielded components such as neurological, fatigue, gastrointestinal, and autoimmune [21]. The same study found that neurological and fatigue symptoms during the initial illness predicted cognitive symptoms. Following this line of research, further analysis of initial COVID-19 symptoms may prove useful in predicting long-term prognosis of Long COVID.

Similar to Long COVID, ME/CFS is difficult to diagnose as there are currently no biological tests to identify it [22]. An estimated 17 million people worldwide [23,24] have this disease, which is characterized by extreme fatigue and the impairment of physical and mental capabilities [25]. Many case definitions and criteria have been developed to aid in the diagnosis of ME/CFS [26–30]. Early detection in patients may be helpful for managing symptoms of ME/CFS due to the severity of the condition. Unfortunately, the lack of biological markers and the difficulty of diagnosis have been limiting factors in this research.

Identifying risk factors in the development of ME/CFS may be helpful in early detection. ME/CFS has been linked to many infectious diseases [31], most notably infectious mononucleosis (IM) [32]. In a study that investigated similarities between Long COVID and ME/CFS [33], considerable overlap occurred between Long COVID and ME/CFS symptomatology. Long COVID symptomatology tended to be less severe than symptoms experienced by participants with ME/CFS. Additionally, most Long COVID symptoms had some improvements over time, except notably for some neurocognitive issues. It should be noted that neurocognitive issues may interfere with a patient's ability to complete lengthy questionnaires.

Completing long questionnaires may be especially draining for those experiencing fatiguing symptoms. The DePaul Symptom Questionnaire (DSQ) [34], used in the current study, has 54 items. Feature reduction could prove useful in reducing the time to finish this questionnaire, for both patients and healthcare providers. In addition, reducing the length of the DSQ could allow for its inclusion in other surveys, potentially allowing for more individuals with ME/CFS to be diagnosed and treated. The current study investigated possible predictive links between initial COVID-19 symptomatology and future development of ME/CFS by analyzing questionnaire responses from the first two weeks of COVID-19 contraction using machine learning models. We also investigated the effects of feature reduction in our predictive models.

## 2. Methods

### 2.1. Data Collection

#### 2.1.1. ME/CFS Sample

The dataset used in this study was collected by staff of the Center for Community Research at DePaul University. Demographic and symptom characteristics of this sample are described elsewhere [33]. Participants were not compensated. In August 2020, IRB permission was obtained to distribute questionnaires online to self-reported COVID-19 long-haulers, individuals who had not fully recovered from a prior COVID-19 infection. The questionnaires were sent out to various social media platforms focused on communication between long-haulers. Participants were asked to complete two symptom questionnaires, one of which detailed "present" symptoms and the other "recalled" symptoms from the first two weeks after COVID-19 diagnosis, an average of 21.7 weeks prior. Data from 299 participants were collected, and the average age was 45 years, with 82% being female, and 86% were white.

#### 2.1.2. The DePaul Symptom Questionnaire (DSQ)

Participants completed the DSQ [34], a 54-item questionnaire that measures ME/CFS symptomatology. Symptoms are measured by frequency and severity on a 5-point Likert scale from 0–4, with lower numbers signifying a lesser frequency/severity of the symptom. Symptoms were averaged and standardized to a 100-point scale to create a symptom composite score. The DSQ has shown a high sensitivity (98%) comparing a physician's diagnosis of ME/CFS and the DSQ's diagnosis [35], as well as yielding clinically useful results [36]. Additionally, the DSQ has demonstrated high test-retest reliability [37], and the DSQ has been able to differentiate between participants with ME/CFS and other chronic illnesses, such as multiple sclerosis [38] and post-polio syndrome [39]. The DSQ may be viewed at this link: https://redcap.is.depaul.edu/surveys/?s=H443P9TPFX (accessed on 20 July 2022).

#### 2.1.3. ME/CFS Diagnosis

Diagnosis of ME/CFS according to the IOM [28] criteria requires a patient to have a substantial reduction or impairment in the ability to engage in pre-illness levels of activity that last for more than 6 months and is accompanied by fatigue. In addition, the patient needs to have the following symptoms: post-exertional malaise, unrefreshing sleep, and either cognitive impairment and/or orthostatic intolerance.

Participant responses to the questionnaire about current symptoms were compared to the IOM ME/CFS case definition criteria [28], and participants were labeled as either meeting the criteria for ME/CFS or not. In this study, 213 (71.2%) participants were labeled as positive for ME/CFS while 86 (28.8%) were labeled as negative for ME/CFS.

#### 2.1.4. Dataset Management

Due to the large imbalance in participants who were labeled with ME/CFS, we used the ROSE package in R to balance the dataset via random oversampling. The "balanced" dataset had 215 (50%) participants labeled as ME/CFS and 215 (50%) as non-CFS partici-

pants. Due to the small size of the original dataset, instead of removing participants with missing values, missing values in the dataset were replaced with $-1$.

### 2.1.5. Statistical Analysis

All analyses were performed using R version 4.2.1 in RStudio. Of the predictive models built using k-nearest neighbors, neural networks, random forests, and logistic regression algorithms, the models built using random forests yielded the most accurate results and are, therefore, the focus of this study. The random forest models were built using the 54 symptom composite scores described above as features on both the "balanced" and "unbalanced" datasets. The random forest algorithm used the parameters ntree = 1500 and mtry = # features in the model. The default value was used for terminal node size and maximum number of terminal nodes. All other parameters were also set to their default values. We performed a leave one out cross-validation procedure to get the accuracy, sensitivity, and specificity of all models. Thirty trials were run for consistency, and t-tests for significance were performed between models.

### 2.1.6. Feature Reduction

The random forest algorithm provided the mean decrease in the Gini coefficient, which measures how each variable contributes to the homogeneity of the nodes. A higher mean decrease in Gini signifies a higher importance of the feature to the model. Using the original dataset and symptom composite scores, we performed a leave one out cross-validation procedure using only the top two features to get the accuracy, sensitivity, and specificity. This process was repeated using three to twenty features to find the model with the highest accuracy. We selected the model with the highest accuracy to obtain the best number of features to use in the future models, as well as the names of these top features. These top features were used to build new models on the balanced dataset. The accuracy, sensitivity, and specificity of these new models were recorded.

A diagram of our process can be seen in Figure 1.

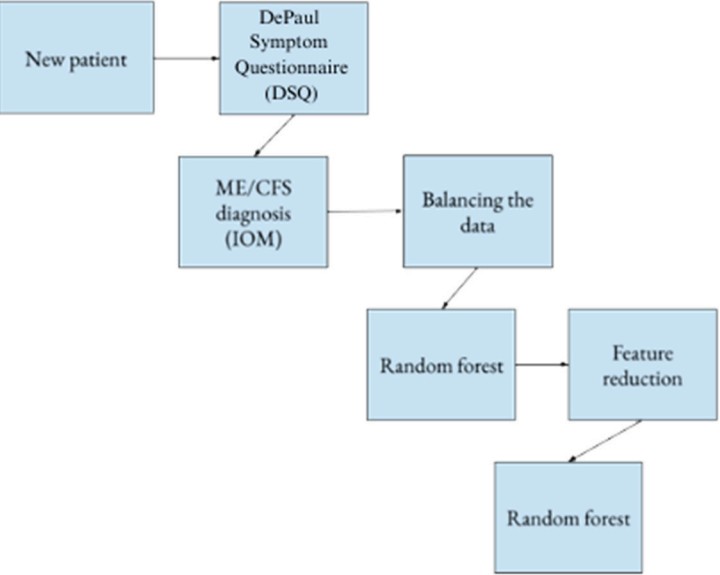

**Figure 1.** Methodology diagram.

### 2.1.7. Tests for Significance

Paired *t*-tests [40] were performed between each of the models to compare the performance of the feature reduced models to the original models. The equation is shown below:

$$t = \frac{|E_1 - E_2|}{\sqrt{q(1-q)\left(\frac{1}{n_1} + \frac{1}{n_2}\right)}}$$

where

$E_1$ = the error rate for model M1;
$E_2$ = the error rate for model M1;
$q = (E_1 + E_2)/2$;
$n_1$ = the number of instances in test set A;
$n_2$ = the number of instances in test set B.

The error rates of our models were computed as the total number of incorrect predictions made by the model divided by the total # predictions. A *t* value $\geq 2$ signifies that we can be 95% confident the difference in the test set performance of the two models is significant.

## 3. Results

Initial results using models built using every symptom on both the "original" and "balanced" datasets are shown below in Table 1.

**Table 1.** Initial results using all 54 features (mean % ± StDev).

| Dataset | Accuracy | Sensitivity | Specificity |
|---------|----------|-------------|-------------|
| Original | 88.26 ± 0.22 | 95.33 ± 0.26 | 70.16 ± 0.30 |
| Balanced | 94.55 ± 0.78 | 91.41 ± 1.08 | 97.69 ± 0.87 |

A plot of the significance of each feature yielded an "elbow" around which we created our feature reduction models. The model using only the "top eight" symptoms predicted ME/CFS development most accurately. The top eight features are shown below in Table 2 along with the corresponding symptom domains.

**Table 2.** Most predictive features.

| # | Feature Name | Domain |
|---|--------------|--------|
| 1 | Fatigue/extreme tiredness | PEM |
| 2 | Mentally tired after the slightest effort | PEM |
| 3 | Feeling unrefreshed after you wake in the morning | Sleep |
| 4 | Minimum exercise makes you physically tired | PEM |
| 5 | Next-day soreness or fatigue after non-strenuous, everyday activities | PEM |
| 6 | Needing to nap daily | Sleep |
| 7 | Dread, heavy feeling after starting to exercise | PEM |
| 8 | Feeling hot or cold for no reason | Neuroendocrine |

The results of the models built using the above top eight features with the "original" and "balanced" datasets are shown below in Table 3.

**Table 3.** Results using top 8 symptoms (mean % ± StDev).

| Dataset | Accuracy | Sensitivity | Specificity |
|---------|----------|-------------|-------------|
| Original | 89.46 ± 0.36 | 95.44 ± 0.35 | 74.17 ± 0.55 |
| Balanced | 93.47 ± 0.99 | 91.13 ± 1.22 | 95.81 ± 1.40 |

In general, the models built on the original 54-item dataset were comparable to the performance of the reduced 8-item models. There were no significant differences between the tested models, as shown in Table 4.

**Table 4.** *t*-test results.

| First Model | Second Model | *t* Value |
|---|---|---|
| Original 54 items | Original 8 items | 0.46 |
| Balanced 54 items | Balanced 8 items | 0.58 |

## 4. Discussion

Our findings indicated that several IOM diagnostic criteria for ME/CFS occurring during the initial weeks after COVID-19 infection predicted Long COVID and the diagnosis of ME/CFS after 6 months. This has implications for aiding those who contract COVID-19 with regard to our understanding of risk factors for developing ME/CFS. For example, if patients recognize the importance of early management of PEM (for example, using pacing), their prognosis might be better, and when patients are not able to use these strategies early in the disease process, there might be a higher chance of developing ME/CFS.

Feature reduction on the models using the eight key symptom scores had statistically insignificant changes in accuracy (per Table 4), suggesting that even this smaller model can accurately predict ME/CFS from initial symptoms. Notably, five of the eight most important features fell primarily in the PEM domain, a finding consistent with previous studies [33,36]. The next most frequent domain was sleep.

However, the dataset does have some limitations, one being that the data may be vulnerable to recall bias or inconsistent memory, as participants were asked to recall symptoms that occurred an average of 21.7 weeks previously. Furthermore, the self-identified long-haulers did not go through a thorough medical examination as the study relied on self-reported data. In addition, the dataset's small size means that any misdiagnoses or inaccurate measurements would have a greater impact on findings.

The dataset was collected from individuals before a COVID-19 vaccination was widely available, which has impacted how symptoms are experienced by individuals [41]. The current availability of the vaccine may mean these same algorithms would produce different results from more recently collected data. Additionally, participant demographics were skewed in many categories, with most participants identifying as female and white. It has been found that people of color tend to be affected more by COVID-19 [42,43]; therefore, the skewed racial distribution of participants might have also influenced the findings.

Future work should include datasets that are larger and more diverse to ensure generalized findings. In addition, more accurate ME/CFS diagnoses performed by a physician may alter or improve results. Additionally, having data that are collected at the time of the initial contraction of COVID-19 instead of using data that were recalled will allow for more accurate results and minimize errors such as recall bias or faulty memory. There is a need for further investigation into why PEM appeared to be the most predictive of ME/CFS.

**Author Contributions:** Conceptualization, C.H., J.S., L.A.J. and J.F.; methodology, C.H., J.S., L.A.J. and J.F.; formal analysis, C.H., J.S., L.A.J. and J.F; writing—review and editing, C.H., J.S., L.A.J., J.F. and D.R.; supervision, J.S. and L.A.J.; funding acquisition, J.S., D.R. and L.A.J. All authors have read and agreed to the published version of the manuscript.

**Funding:** This work was supported by the National Science Foundation award 1950894 and the National Institute of Neurological Disorders and Stroke (Grant number 5R01NS111105).

**Institutional Review Board Statement:** All data were collected with approval from the DePaul Institutional Review Board.

**Informed Consent Statement:** Not applicable.

**Data Availability Statement:** Data can be made available by contacting the first author.

**Conflicts of Interest:** The authors declare no conflict of interest.

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
