# Peer review of "Predicting Myalgic Encephalomyelitis/Chronic Fatigue Syndrome from Early Symptoms of COVID-19 Infection"

_psych, doi:10.3390/psych5040073_

Round 1

Reviewer 1 Report

I thank for the opportunity to review this manuscript. The authors raise an important question: If we could identify patients with a high risk for developing long-Covid already when they get sick with the primary Covid-19 infection, active measures taken early enough could perhaps reduce that risk. The method they use is not ideal (to recall early symtoms afterwards) but the best that can be done at this point. The authors have interesting data to report, but in its current form the manuscript fails to forward these data in a readerfriendly manner. So I encourage the authors to keep on working on the manuscript to deliver more of their data that has clinical and scientific interest.

Some suggestions for improving the manuscript:

1) The introduction is (too) long (for a short report). I understand that the authors want to introduce the reader to much of the not so well known disease (ME/CFS). Please refer to review articles and some points could be moved to discussion.

2) “Participants were asked to complete two symptom questionnaires, one of which detailed present symptoms and the other recalled symptoms from the first two weeks after COVID-19 diagnosis–an average of 21.7 weeks prior.”

The most predictive 8 features are presented in Table 3. I and probable also those readers unfamiliar with DePaulQuestionnaire would appreciate the list of all the features / domains that were initially entered into the tested models- Are they too many to be presented? I would find it as interesting to see, which features were least predictive. This could be more informative than your current Figure 1. 

3) The data treatment for the modelling is complex and beyond my expertize. It recommend that this part of the manusript would be reviewed by a mathematician. 

Minor suggestions:

“Similar to long COVID, without a biomarker or a diagnostic test the diagnosis of ME/CFS challenging [22].” Diagnostic criteria that include PEM (post-exertional malaise) as an obligatory feature define the condition most accurately [29], [30].

“participant demographics were skewed in many categories”. The participants represented a typical gender distribution of strong female predominance (81.6%).  [question: how did this skewness invalidate the model?]

Although the manuscript makes itself understandable, there is room language revision. 

Author Response

Reviewer 1:

“I thank for the opportunity to review this manuscript. The authors raise an important question: If we could identify patients with a high risk for developing long-Covid already when they get sick with the primary COVID-19 infection, active measures taken early enough could perhaps reduce that risk. The method they use is not ideal (to recall early symptoms afterward) but the best that can be done at this point. The authors have interesting data to report, but in its current form, the manuscript fails to forward these data in a reader-friendly manner. So I encourage the authors to keep on working on the manuscript to deliver more of their data that has clinical and scientific interest.”

            We thank this reviewer for feeling that our paper raises important questions, and we now have tried to present the data in a more user-friendly way, and have also incorporated suggestions below as well as that of the other reviewers.

Reviewer 1 felt that the introduction was too long for a short report.  We have now referred to review articles but our feeling is that readers unfamiliar with ME/CFS or Long COVID will need this framing to understand the importance of our article, which as found for the first time that during the first few weeks, it is possible to predict who will later have more problems, and unless readers can see other efforts in this area, they would not be able to understand the significance of our findings. And we do believe that this article will be widely read around the world for its unique findings.

“The most predictive 8 features are presented in Table 3. I and probable also those readers unfamiliar with DePaul Questionnaire would appreciate the list of all the features / domains that were initially entered into the tested models.”

            These items are widely available in our other publications and as there are 54 items, would prefer not to list them all here. Readers interested in the demographic and symptom characteristics are provided reference to another article where they are listed. However, if that is something that you would prefer, we would of course be willing to list them. Also, the current Figure does provide readers with a simple way of understanding the method, and as we are trying to be user friendly, we would prefer to keep this figure, but of course, if you prefer we eliminate it, we would be willing to do so. 

Reviewer 1 also stated that “The data treatment for the modeling is complex and beyond my expertize. It recommends that this part of the manuscript would be reviewed by a mathematician.”  Our authors have expertise in computer science and statistics, so we feel rather confident about our findings, and it does appear that the other reviewers were OK with our analyses, so we will keep them as they are.  We feel that this is one of the real contributions of this paper, that being, sophisticated analyses, which the Long COVID field has needed.

Finally, Reviewer 1 stated that “Although the manuscript makes itself understandable, there is room language revision.”

            We agree with this statement and have edited the paper to make is more understandable and clean up some language issues that were noticed by this reviewer.

“Similar to long COVID, without a biomarker or a diagnostic test the diagnosis of ME/CFS challenging [22].”

“Diagnostic criteria that include PEM (post-exertional malaise) as an obligatory feature define the condition most accurately [29], [30].

            We have fixed these grammar problems.

“participant demographics were skewed in many categories”. The participants represented a typical gender distribution of strong female predominance (81.6%).  [question: how did this skewness invalidate the model?]”

            We do not believe that this skewness invalidated our model but we just report it here so the reader does know that this study and others have had an overrepresentation of females, as females do tend to have more auto-immune disorders than men.

Reviewer 2 Report

This is a paper about the relationship between long covid and ME/CFS. Previous findings show that long covid is characterized by fatigue and a part of the patients do not recover from illness and become a condition called ME/CFS, a intractable serious chronic illness with no established biomarker and treatment. The difference between 'recovered' and 'non-recovered' is an important issue in the context of long COVID but still remained elusive. In this sense, this study deals with an important clinical question.

The methodology seems satisfactory with good numbers and using well-established questionnaire (DSQ)and the obtained data is trustworthy. The conclusion - stressing the importance of post exertional malaise (PEM) is logical, because this is the key feature of ME/CFS. 

The paper also mentions several critical limitations such as recall bias.

As a reader of this paper, I am interested in the following point, which may be mentioned in this paper.

Which additional factor(s) influence the transition to ME/CFS, if a patient have PEM in the first two weeks?  Intuitively, if such patients recognize the importance of management to PEM (such as pacing, chance to get help from others, etc), the prognosis might change. But if they failed to do so, there would be more chance to transit to ME/CFS.

Author Response

Reviewer 2:

This is a paper about the relationship between long covid and ME/CFS. Previous findings show that long covid is characterized by fatigue and a part of the patients do not recover from illness and become a condition called ME/CFS, a intractable serious chronic illness with no established biomarker and treatment. The difference between 'recovered' and 'non-recovered' is an important issue in the context of long COVID but still remained elusive. In this sense, this study deals with an important clinical question.

            We appreciate the positive comments that this reviewer mentioned regarding our paper.

The methodology seems satisfactory with good numbers and using well-established questionnaire (DSQ)and the obtained data is trustworthy. The conclusion - stressing the importance of post exertional malaise (PEM) is logical because this is the key feature of ME/CFS.

            We again appreciate the positive comments from this reviewer.

The paper also mentions several critical limitations such as recall bias.

            We agree that we do mention limitations.

As a reader of this paper, I am interested in the following point, which may be mentioned in this paper. Which additional factor(s) influence the transition to ME/CFS, if a patient have PEM in the first two weeks?  Intuitively, if such patients recognize the importance of management to PEM (such as pacing, chance to get help from others, etc), the prognosis might change. But if they failed to do so, there would be more chance to transit to ME/CFS.  

            This is a great point made by the reviewer, and we agree that if such patients recognize the importance to management of PEM, such as pacing, the prognosis might be better, and if they are not able to use these strategies, there might be a higher chance to develop ME/CFS. We have now added this excellent point to the paper.

Reviewer 3 Report

Long COVID (LC) does indeed present with a significant challenge both to healthcare providers and those who experience it. Over time, it is perceived that the impairments will persist over time useless interventions are considered and the condition is pervasive.  Overall, this is a very interesting and important paper. The authors clearly address the limitations of the study such as self-diagnosis of long COVID and the imbalance of sex and ethnicity. The methodology and analysis are sound and the findings presented in a logical manner.

One thing that I am concerned about is the link between CFS/ME and LC. I think it would be better to use the term post-viral fatigue syndrome as CFS/ME is still of unknown aetiology. Another consideration is the length of symptom duration when comparing LC to CFS/ME.

Author Response

Reviewer 3

Long COVID (LC) does indeed present with a significant challenge both to healthcare providers and those who experience it. Over time, it is perceived that the impairments will persist over time useless interventions are considered and the condition is pervasive.  Overall, this is a very interesting and important paper. The authors clearly address the limitations of the study such as self-diagnosis of long COVID and the imbalance of sex and ethnicity. The methodology and analysis are sound and the findings presented in a logical manner.

            We appreciate the positive comments of Reviewer 3.

One thing that I am concerned about is the link between CFS/ME and LC. I think it would be better to use the term post-viral fatigue syndrome as CFS/ME is still of unknown aetiology. Another consideration is the length of symptom duration when comparing LC to CFS/ME.

            We agree that ME/CFS often has an unknown aetiology, but many cases do develop after Mono and exposure to Epstein Barr virus. Long COVID has some difficulties with its case definition as well. As this paper deals with ME/CFS and a particular case definition, if we were to call this disease something more general like post-viral fatigue syndrome, it might be harder for us to keep the focus on ME/CFS and a very specific case definition that we are using.  We also agree that symptom duration for ME/CFS will be longer than Long COVID, as those with ME/CFS have had this disease for a longer period of time, and those with Long COVID could only have been sick for the last few years. Still,  comparing these two illnesses might have a number of important contributions for researchers, as we argue in our paper.

Round 2

Reviewer 1 Report

Thank you for your comment and improvements. I still have three suggestions, which I strongly wish you to consider for the final version.

1) Figure 1: It would help the reader if DSQ in the table box is opened. There is enough room for the longer version:

DePaul Symptom Questionnaire (DSQ)

2) When I look at the 8 predictive features, to me it looks as your data reduction resulted very close to the IOM criteria for ME/CFS (of course exluding the item of the 6 month duration). If my interpretation is right, I would encourage you to make a stronger conclusion by finishing your abstract and also including in your first chapitre of your discussion (as your main finding) with something like: "Presence of most of the IOM diagnostic criteria for ME/CFS already during the initial weeks with Covid-19 infection predict long-Covid and the diagnosis of ME/CFS after 6 months."  

3) Is your title completely right?

"Predicting Myalgic Encephalomyelitis/Chronic Fatigue Syndrome From Long-COVID Symptoms"

To me, the correct title would be:

"Predicting Myalgic Encephalomyelitis/Chronic Fatigue Syndrome from early symtoms of COVID-19 infection"

I congratulate you for a nice article.

Author Response

Manuscript ID: psych-2619171

Type of manuscript: Brief Report

Title: Predicting Myalgic Encephalomyelitis/Chronic Fatigue Syndrome From Long-COVID Symptoms

Dear Ms. Marta Spasic:

We were asked to respond to a few comments from Reviewer 1, and our responses are below:

We appreciate this reviewer writing:  “I congratulate you for a nice article.”

Reviewer 1 wrote that for Figure 1: It would help the reader if DSQ in the table box is opened. There is enough room for the longer version: DePaul Symptom Questionnaire (DSQ)

              We agree and have now changed this.

2) When I look at the 8 predictive features, to me it looks as your data reduction resulted very close to the IOM criteria for ME/CFS (of course excluding the item of the 6 month duration). If my interpretation is right, I would encourage you to make a stronger conclusion by finishing your abstract and also including in your first chapter of your discussion (as your main finding) with something like: "Presence of most of the IOM diagnostic criteria for ME/CFS already during the initial weeks with Covid-19 infection predict long-Covid and the diagnosis of ME/CFS after 6 months."  

              We agree and now have inserted this into the discussion.

3) Is your title completely right? "Predicting Myalgic Encephalomyelitis/Chronic Fatigue Syndrome From Long-COVID Symptoms" To me, the correct title would be: "Predicting Myalgic Encephalomyelitis/Chronic Fatigue Syndrome from early symptoms of COVID-19 infection"

              We agree and the title has been changed.

Sincerely,

Len Jason
